# New Onset Autoimmune Diseases after the Sputnik Vaccine

**DOI:** 10.3390/biomedicines11071898

**Published:** 2023-07-04

**Authors:** Olga Vera-Lastra, Gabriela Mora, Abihai Lucas-Hernández, Alberto Ordinola-Navarro, Emmanuel Rodríguez-Chávez, Ana Lilia Peralta-Amaro, Gabriela Medina, María Pilar Cruz-Dominguez, Luis J. Jara, Yehuda Shoenfeld

**Affiliations:** 1Internal Medicine Department, Hospital de Especialidades, Dr. Antonio Fraga Mouret, Centro Médico Nacional La Raza, Instituto Mexicano del Seguro Social (IMSS), Mexico City 02990, Mexico; ranitaper22@gmail.com; 2Inmunology Department, Hospital Militar Central, Cirujano Mayor Dr. Cosme Argerich, Buenos Aires C1426, Argentina; serv.inmuno@hmc.mil.ar; 3Rheumatology Department, Centro Médico Nacional 20 de Noviembre, Instituto de Seguridad y Servicios Sociales de los Trabajadores del Estado (ISSSTE), Mexico City 03104, Mexico; abihai.lucas1991@gmail.com; 4Infectious Diseases Department, Instituto Nacional de Ciencias Médicas y Nutrición Salvador Zubirán, Mexico City 14080, Mexico; albertoordinolamd@gmail.com; 5Neurology Department, Hospital de Especialidades, Dr. Antonio Fraga Mouret, Centro Médico Nacional La Raza, Instituto Mexicano del Seguro Social (IMSS), Mexico City 02990, Mexico; e2manuel_rch@hotmail.com; 6Translational Research Unit, Hospital de Especialidades, Dr. Antonio Fraga Mouret, Centro Médico Nacional La Raza, Instituto Mexicano del Seguro Social (IMSS), Mexico City 02990, Mexico; dragabymedina@yahoo.com.mx; 7Direction of Research and Education, Hospital de Especialidades, Dr. Antonio Fraga Mouret, Centro Médico Nacional La Raza, Instituto Mexicano del Seguro Social (IMSS), Mexico City 02990, Mexico; drapilarcd@gmail.com; 8Rheumatology Division, Instituto Nacional de Rehabilitación “Luis Guillermo Ibarra Ibarra”, Mexico City 14389, Mexico; luis_jara_quezada@hotmail.com; 9Zabludowicz Center for Autoimmune Diseases, Sheba Medical Center, Sackler Faculty of Medicine, Tel Aviv University, Tel-Hashomer, Ramat Gan 52621, Tel Aviv 69978, Israel; yehuda.shoenfeld@sheba.health.gov.il

**Keywords:** COVID-19 vaccines, sputnik vaccines, systemic autoimmune diseases, adverse events, Autoimmune/Inflammatory Syndrome Induced by Adjuvants (ASIA)

## Abstract

The vertiginous advance for identifying the genomic sequence of SARS-CoV-2 allowed the development of a vaccine including mRNA-based vaccines, inactivated viruses, protein subunits, and adenoviral vaccines such as Sputnik. This study aims to report on autoimmune disease manifestations that occurred following COVID-19 Sputnik vaccination. Patients and Methods: A retrospective study was conducted on patients with new-onset autoimmune diseases induced by a post-COVID-19 vaccine between March 2021 and December 2022, in two referral hospitals in Mexico City and Argentina. The study evaluated patients who received the Sputnik vaccine and developed recent-onset autoimmune diseases. Results: Twenty-eight patients developed recent-onset autoimmune diseases after Sputnik vaccine. The median age was 56.9 ± 21.7 years, with 14 females and 14 males. The autoimmune diseases observed were neurological in 13 patients (46%), hematological autoimmune manifestations occurred in 12 patients (42%), with thrombotic disease observed in 10 patients (28%), and autoimmune hemolytic anemia in two patients (7.1%). Rheumatological disorders were present in two patients (7.1%), and endocrine disorders in one patient (3.5%). Principio del formulario Conclusion: Although the COVID-19 Sputnik vaccine is generally safe, it can lead to adverse effects. Thrombosis and Guillain-Barre were the most frequent manifestations observed in our group of patients.

## 1. Introduction

Once the COVID-19 pandemic was declared in 2020, the entire world anticipated the development of a vaccine capable of saving numerous lives globally to diminish mortality. Extensive trials were conducted to ensure the safety of these vaccines and to address any adverse events that may arise [1]. However, the literature began to document adverse effects associated with COVID-19 vaccines, albeit in a minority of individuals, including neurological, cardiovascular, cerebrovascular, and autoimmune syndromes among others [2]. A recent systematic review of the literature revealed an increase in COVID-19 vaccine acceptance rates, therefore the number of people exposed to the vaccine also increased, increasing the number of vaccinated people. Consequently, vaccination programs should provide comprehensive information regarding the COVID-19 vaccine, its potential side effects, and the overall impact of the pandemic [3].

In February 2021, the Gamaleya National Research Centre for Epidemiology and Microbiology in Moscow, Russia published their phase 3 results for the rAd26 and rAd5 vector-based vaccine “Sputnik”. This vaccine utilizes rAd type 26 and rAd type 5, both carrying the gene for the viral glycoprotein S. It demonstrated efficacy comparable to other globally approved vaccines [4]. Like any vaccine, some secondary effects have been reported; however, in most cases, these effects have been mild [5,6,7].

Recent reports have surfaced regarding cases of autoimmune diseases following the administration of the Sputnik vaccine. These include reactive arthritis, thrombocytopenia, thrombosis, severe eczema, and neurological manifestations such as Bell’s palsy and demyelinating diseases or relapses of multiple sclerosis, among others [8,9,10,11,12]. The mechanisms underlying the development of adverse effects like thrombosis due to the Sputnik vaccine remain unclear but may be similar to those proposed for the AstraZeneca vaccine (AstraZeneca: AZ, Cambridge, UK). These mechanisms involve the presence of antibodies against platelet factor 4 (PF4) and the presence of ethylenediaminetetraacetic acid (EDTA) in both Sputnik and AstraZeneca vaccines, which predispose to clot formation, neutrophil activation, and the formation of neutrophil extracellular traps (NETs) [13,14]. This study aims to describe the newly emerged autoimmune syndromes following the administration of the Sputnik vaccine against COVID-19 in two countries, Mexico, and Argentina, within two referral hospitals.

## 2. Patients and Methods

We conducted a retrospective study of patients with new-onset autoimmune diseases occurring after COVID-19 vaccine inoculation between March 2021 and December 2022, at two referral hospitals: one in Mexico City and the other in Buenos Aires, Argentina. Our study focused on patients who had received the Sputnik vaccine and subsequently developed recent-onset autoimmune diseases.

We assessed all patients for clinical manifestations and autoantibodies associated with autoimmune diseases following COVID-19 vaccination. Additionally, we recorded the time of appearance of autoimmune manifestations after vaccine administration. The inclusion criteria consisted of patients who met the criteria for a new-onset autoimmune disease with a production of antibodies that fulfilled the specific diagnostic/classification criteria and/or nomenclature for each rheumatic/neurology autoimmune disease exhibiting a temporal association (≤90 days) with COVID-19 vaccination, without any other reported triggers, and who were older than 16 years of age. Informed consent was obtained from all participants. Patients with pre-existing autoimmune diseases, those who had received more than two vaccinations, and individuals who had received other COVID-19 vaccines such as Comirnaty (Pfizer-BioNTech., New York, NY, USA), Covishield (AstraZeneca: AZ, Cambridge, UK), Coronavac (Sinovac Biotech, Beijing, China), Convidecia (CanSino Biologics, Tianjin, China), Covaxin (Bharat Biotech, Hyderabad, India), Jcovden (Janssen, Leiden, The Netherlands), Spikevax (Moderna Inc., Cambridge, MA, USA), and BBIBP-CorV (Sinopharm, Beijing, China) were excluded from the study. Blood cell count and blood chemistry tests were performed on all patients, and a negative PCR test for COVID-19 was confirmed. Demographic, clinical, and laboratory data were collected starting from the first day after vaccination.

For patients with autoimmune rheumatological manifestations, we measured antinuclear antibodies (ANA), anti-DNA antibodies, antibodies extracted from the nucleus (ENA), antineutrophil cytoplasmic antibodies (ANCA), rheumatoid factor (RF), anti-cyclic citrullinated peptide (anti-CCP), and ferritin. For patients with endocrinological manifestations, we evaluated the thyroid profile, anti-TSH receptor antibodies (anti-TSH-R), thyroid-stimulating immunoglobulin (TSI), anti-peroxidase antibodies (TPO), and antithyroglobulin (anti-Tg). For neurological patients, we tested for antibodies to the acetylcholine receptor (AChR), anti-aquaporin-4 (AQP4), antibodies against neuronal surface receptors, including anti-N-methyl-D-aspartate (anti-NMDAr) and anti-glutamic acid decarboxylase antibody (anti-GABA), as well as anti-glutamic acid decarboxylase antibodies (GADAb). These autoantibodies were analyzed in cerebrospinal fluid studies (CSF). Brain, cervical, thoracic, and lumbar spine magnetic resonance imaging (MRI) were also performed. The laboratory tests were performed using the following methods: ANA were tested using indirect immunofluorescence (IFI) on a monolayer of human larynx epithelioma cancer (HEp-2) cells. ANCA were tested using IFI on ethanol-fixed human neutrophils. Enzyme-Linked ImmunoSorbent Assay (ELISA) was used to test ENA, anti-DNA, RF, anti-CCP, anti-TSH-R, anti-TPO), anti-Tg, NMDAr, and anti-GABA, GADAbn. Immunofluorescence (IF) was used to test for TSI. Chemiluminescence was used for thyroid function testing and ferritin. AQP4 were tested by incubating muscle sections from frozen mouse muscles with the patient’s cerebrospinal fluid at different dilutions (1:10, 1:100) for 24 h at 4 °C. The sections were then labeled with secondary antibodies specific to human immunoglobulin G (IgG) labeled with DayLight (1:200) from Inmunoreagents (Sta. Cruz lab). Hoescht staining (from InvitroGen) was performed as a counterstain, and the sections were mounted with Vectashield (fluorescence mounting medium; Vector Labs).

The collected information was organized into tables for subsequent analysis. We employed descriptive statistics, frequencies, percentages, mean, and standard deviation according to the distribution.

## 3. Results

Our study observed a total of 123 post-vaccine effects, out of which 28 occurred following the administration of the Sputnik vaccine. The median age of the patients was 56.9 ± 21.7 years, with an equal distribution of females (14) and males (14). Most patients were generally healthy individuals, except for two cases of Guillain-Barre syndrome (GBS) in individuals who had a history of arterial hypertension (AH).

Within the subgroup of patients with hematological conditions, frequent comorbidities included cancer and AH. Most patients (57%) experienced clinical manifestations after receiving the second dose of the vaccine. The primary autoimmune diseases observed were of neurological origin, including Guillain-Barre syndrome, neuromyelitis optica spectrum disorder (NMOSD), myasthenia gravis, chronic immune demyelinating polyneuropathy, and opsoclonus-myoclonus-ataxia syndrome. Hematological conditions included deep vein thrombosis (DVT) of the lower extremities, pulmonary thromboembolism, multiple venous thrombosis, and autoimmune hemolytic anemia. Rheumatological conditions observed consisted of dermatomyositis and adult-onset Still’s disease. Additionally, there was a case of endocrine manifestation with Graves’ disease. Four of these patients presented Antinuclear antibodies (two patients with Guillain-Barre syndrome, one transverse myelitis, and one with dermatomyositis). The two patients with NMOSD had positive AQP-4 antibodies (Figure 1), and the patients with Myasthenia Gravis had antibodies against the receptor for acetylcholine.

Figure 2 shows Magnetic Resonance Imaging (MRI) in two patients with NMOSD. The cerebrospinal fluid analysis revealed inflammatory fluid with pleocytosis and cytological albumin dissociation in patients with Guillain-Barré syndrome. Furthermore, nerve conduction velocities showed prolonged latencies in both sensory and motor components. Unfortunately, the patients who experienced thrombosis did not have antibodies tested for antiphospholipid syndrome.

Table 1 provides a summary of the main characteristics of patients who developed autoimmune diseases following inoculation with the COVID-19 Sputnik vaccine.

## 4. Discussion

This study found that the autoimmune diseases associated with the COVID-19 Sputnik vaccine were of neurological nature, followed by hematological, rheumatological, and endocrine manifestations. These adverse effects occurred in most cases after the second dose. Affected patients responded adequately to established medical treatment, with steroids being the cornerstone in all of them, plus other treatments such as rituximab in specific cases such as NMOSD. It is emphasized that neurological manifestations (in 13 out of 28 patients) were the most frequently associated with Sputnik COVID-19. Post-vaccine effects led to Guillain-Barre syndrome in six out of 13 neurological cases, and thrombotic diseases occurred in 10 out of 28 patients with hematological complications.

In our study, we found 10 cases of thrombotic events after vaccination. Similar findings have been documented in a study by Vallone et al. [15] after the first dose of adenoviral-based COVID-19 vaccines, with no significant differences regarding the type of vaccine inoculated. Other authors reported a predominance of thromboembolic events and vaccine-induced immune thrombocytopenia and thrombosis (VITT) after the application of the A-Z vaccine in a systematic review [16]. These infrequent side effects resemble those of critical SARS-CoV-2 infection, with systemic immunothrombosis affecting several territories [17,18]. While natural SARS-CoV-2 infection is mediated by antigens like single-stranded RNA, vaccine-associated immunothrombosis is related to DNA adenovirus-vectored platforms [19]. Pathogenic mechanisms proposed for these events are: (1) as an adjuvant, double-stranded, negatively charged DNA forms complexes with platelet-factor 4 (PF4), amplifying TLR9-plasmacytoid dendritic cell interferon-alpha (IFN-α) production, leading to autoantibody formation and thrombosis in VITT; (2) occurrence of local tissue microtrauma, at the site of inoculation, with local microbleeding and immune cell activity, bringing the vector adenoviral DNA into contact with PF4, both taken up by dendritic cells, thereafter, memory B cell engagement in the regional lymph nodes and PF4 autoantibody production; (3) molecular mimicry between spike protein and PF4 epitopes [13]; (4) occurrence of DNA-PF4 complex interactions with other vaccine contents, like protein contaminants, the number of spike protein DNA copies, charged agents like EDTA [20]; and, finally, (5) release of self-DNA by neutrophil extracellular traps (NETs) at the site of injection/injury; the latter mechanism is common in severe COVID-19 disease. Self-DNA released by NETs is citrullinated and alters DNA-protein interactions through electric rebalancing [21].

We also found acute adverse neurological effects in 13 patients, accounting for most of the patients studied in this group. Neurological manifestations have been reported after inoculation with other vaccines during this pandemic, such as mRNA vaccines, and also after the A-Z vaccine [11,22]. The explanation for these neurological events has been that antibodies against the S1 domain of the SARS-CoV-2 spike protein used in every vaccine have been found in the cerebrospinal fluid of affected patients [23]. It has also been hypothesized that molecular mimicry between the vaccine components and the cerebral or spinal cord tissues triggers autoimmune responses [24].

Our study observed two cases of new-onset rheumatological diseases (Dermatomyositis and adult-onset Still’s disease). Myopathies have been reported not only after COVID-19 vaccination, but also after Hepatitis B, influenza, smallpox, tetanus, and other vaccines [25,26,27]. Several mechanisms have been proposed for the induction of autoimmunity in these patients, including molecular mimicry, epitope spreading, bystander activation, or superantigenic T-cell activation. Although the exact mechanism by which these vaccines induce autoimmune disease is not yet known, we have seen an increasing number of reports of arthralgias and arthritis after COVID-19 vaccination in recent years, which have also been explained by molecular mimicry [28,29].

Interestingly, some patients exhibited ANAs, which are defining features of autoimmune connective tissue disease. ANAs have also been observed in relation to other autoimmune diseases, such as neurological and endocrinological conditions, among others. An interesting finding is that our patient with GBS presented seropositivity for ANA. This is an infrequent presentation, as a positive ANA has been found in only a few cases of GBS, such as Miller-Fisher Syndrome, which is a variant of GBS. The explanations for this finding may be multifactorial, suggesting a manifestation of autoimmunity triggered by a viral agent since this syndrome is often preceded by a mild viral or bacterial infection, typically respiratory or gastrointestinal in nature [30]. Regarding the treatment of GBS, the use of intravenous immunoglobulin (IVIG) can be associated with elevated ANA levels, although in our patients the ANAs were tested prior to the use of IVIG and this may be a coincidental finding [31]. Recently, the induction of ANA, ENA, and autoimmune diseases, such as lupus cerebritis, autoimmune hepatitis, and glomerulonephritis, among others, has been reported after COVID-19 vaccination [32,33,34]. However, according to a study, no ANA induction or increase was observed after administering the second dose of BNT162b2 and mRNA-1273 vaccines against SARS-CoV-2, six months after vaccination. Another study also found no association between BNT162b2 vaccine administration and changes in antiphospholipid syndrome (APS) and ANA titers [35,36]

The effectiveness of the Sputnik vaccine has been proven since its first report in 2021, with an efficacy of 91.6% [1]. Real-life experiences have also demonstrated their effectiveness in large populations, as reported by Bello-Chavolla et al. [37]. They found that out of 793,487 people vaccinated in Mexico, 79,551 received the Sputnik vaccine, and its effectiveness was calculated at 67.73% with the first dose, which increased to 78.75% with the second dose. Additionally, this vaccine has shown effectiveness against different SARS-CoV-2 variants of concern, with virus-neutralizing activity (VNA) like that of the original variant [38]. Moreover, the study conducted by Erra et al. [5] on patients with inborn errors of immunity found that they achieved seroconversion.

There is no doubt about the effectiveness of the Sputnik vaccine in people with different backgrounds. However, it has been reported to have more side effects than other vaccines in some cases. For instance, in a study conducted in Jordan, where 658,428 people were vaccinated with four different types of vaccines, all Adverse Events Following Immunization (AEFI) were reported. Among the vaccinated individuals, 1390 people received the Sputnik vaccine, and the overall incidence of AEFI for this vaccine was 50.8%, which is significantly higher than the overall incidence of AEFI for the other vaccines (BNT162b2: 32.6%, BBIBP-CorV: 16%, ChAdOx1 nCoV-19: 37.4%). The overall rates of systemic, local, and immediate hypersensitivity to AEFIs with Sputnik were 24.9%, 27.5%, and 1.4%, respectively [39]. In another study conducted in Argentina on patients with rheumatic diseases, 1234 patients received different vaccines, and the incidence of adverse events for Sputnik V was 195.4/1000, compared to 359/1000 for ChAdOx1 nCOV-19 [40]. In most cases, the adverse events following immunization with Sputnik V were mild, such as fever or pain at the injection site [40]; however, some cases reported other effects. Table 2 presents a compilation of reported cases in the literature regarding adverse effects associated with the Sputnik vaccine [41,42,43,44,45]. Nevertheless, we found autoimmune events that are not commonly associated with this vaccine.

Sputnik V is a COVID-19 adenoviral vector vaccine. While this vaccine is relatively safe, it is not without risks and can have adverse effects. One such effect is VITT, which has been described for both the Sputnik and A-Z vaccines, as they use adenoviral vectors with similar mechanisms for thrombosis [9,10].

When human peripheral blood mononuclear cells (PBMC) are exposed to Ad5 and Ad26, they elicit type-I and type-II IFN responses, along with the induction of certain cytokines and chemokines. Innate immune system cells tend to release pro-inflammatory signals in response [46]. In individuals vaccinated with the A-Z vaccine, there is a correlation among toll-like receptor (TLR) induced B-cell activation, natural killer cell activation, monocyte activation, and SARS-CoV-2-neutralizing antibody titers. However, this response appears to be diminished in older subjects, as age-related decreases in TLR function can negatively impact immunogenicity [47]. The influx and activation of innate immune cells at the site of inoculation may trigger the activation of pattern recognition receptors, creating a pro-inflammatory environment that leads to specific reactogenicity, both locally and systemically [48]. These reactions may explain the cytokine responses associated with systemic adverse events following adenoviral-based vaccinations, rather than being directly related to humoral immune responses.

It is worth noting that, besides a single systematic review reporting one case of neurological affection after the Sputnik vaccine, there have been reports of ChAdOx1 nCoV-19-associated transverse myelitis occurring around days 10–14 after vaccination, with poor immune response. This suggests that the depletion of CD20 B cells may delay the production of anti-SARS-CoV-2 antibodies [42]. Although these events typically occur after the second dose, there have been cases, especially in genetically predisposed individuals, where they occur even after the first vaccine, as reported for some other adenovirus-based vaccines. However, following a booster dose, an enhanced immune response is expected, which can potentially lead to immunological events.

As in the SARS-CoV-2 infection, which triggers various autoimmune diseases [45], some authors have recently reported adverse effects such as exacerbation [49] or the recent appearance of new autoimmune diseases, probably in individuals genetically predisposed to different anti-COVID-19 vaccines, especially Pfizer, AstraZeneca, and Sputnik vaccines. This occurs through mechanisms such as molecular mimicry and others [50]. Lavi T et al. [51] reported deregulation in the titers of several autoantibodies against neuronal and CNS-related autoantigens in convalescent patients from COVID-19. More information is needed to understand the link between these neural autoantibodies and autoimmune neurological diseases in patients with COVID-19 and those induced by anti-COVID-19 vaccines.

A better understanding of the pathogenesis of autoimmune diseases induced by anti-COVID-19 vaccines will allow for better therapeutic strategies, such as IVIG and anti-cytokine therapy [52]. Although adverse events from autoimmune diseases, particularly neurological and hematological, associated with the Sputnik vaccine were rare in our study, further research is warranted to investigate these phenomena.

Our study has the limitation of a small patient group. Immunological studies were not conducted in patients with thrombosis and, unfortunately, we did not perform antiganglioside antibody testing in patients with GBS. However, its strength lies in the scarcity of studies investigating adverse autoimmune effects associated with the Sputnik vaccine, and our report includes a larger number of patients with diverse clinical manifestations.

## 5. Conclusions

The COVID-19 Sputnik vaccine is considered safe; however, it can lead to adverse effects in a minority of patients, including neurological, hematological, rheumatological, and endocrinological autoimmune diseases. Notably, thrombosis and Guillain-Barre syndrome were the most frequently observed conditions not previously reported in association with this vaccine. Our patients presented a range of clinical manifestations of autoimmune diseases and met the criteria for Autoimmune/Inflammatory Syndrome Induced by Adjuvants (ASIA) caused by COVID-19 Sputnik vaccines. All the secondary effects reported here responded well to the standard treatment for each disease.

Healthcare providers should be mindful of these potential adverse effects and closely monitor individuals who receive the COVID-19 Sputnik vaccine for any sign or symptom of an autoimmune disease. Further research is necessary to investigate the possible connection between COVID-19 Sputnik vaccination and autoimmune diseases.

## Figures and Tables

**Figure 1 biomedicines-11-01898-f001:**
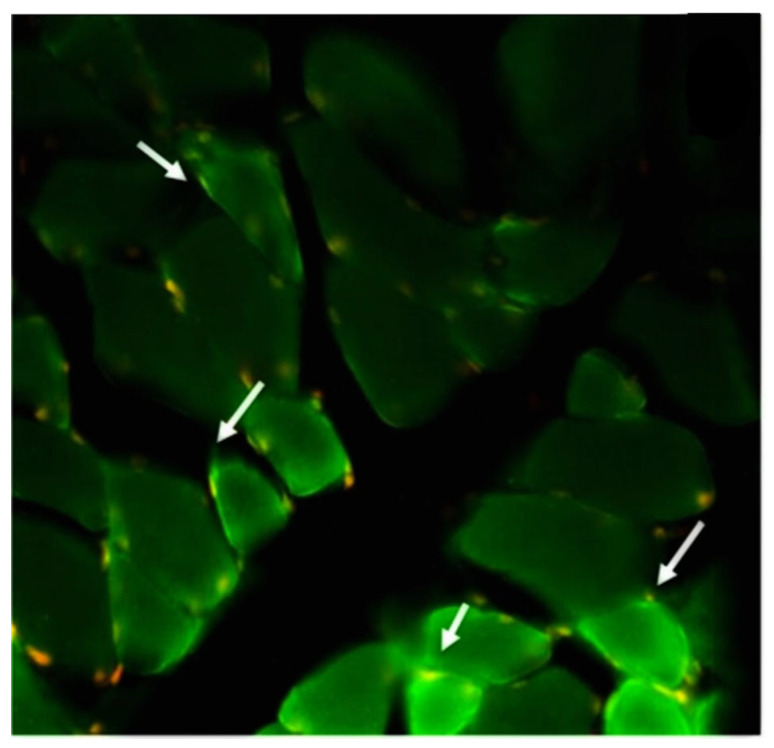
An intense reaction is observed in the patient’s cerebrospinal fluid autoIgGs against proteins located in the muscle sarcolemma, corresponding to aquaporins (green dots; white arrow) and red/yellow dots representing muscle cell nuclei.

**Figure 2 biomedicines-11-01898-f002:**
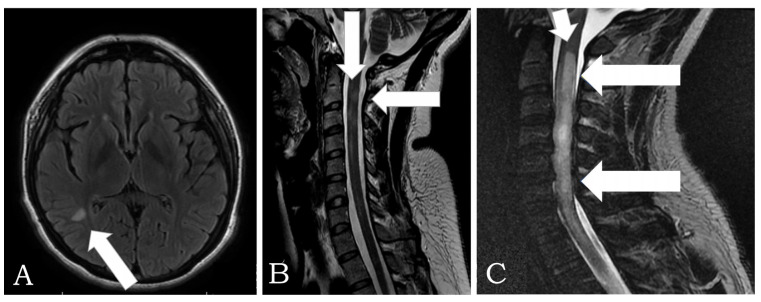
(**A**) MRI in FLAIR, axial plane: a hyperintense lesion is observed in the parieto-occipital region with an ovoid shape of 6 mm, in relation to a single lesion with demyelinating characteristics in a patient with Neuromyelitis Optica Spectrum Disorder (NMOSD). (**B**) MRI is observed in the T2-weighted cervical spine, sagittal plane: multifocal lesion with a tendency to an extensive longitudinal pattern, with involvement of segments C2, C3, C4, and C7, in relation to myelitis with inflammatory characteristics due to NMOSD of previous patients with a brain lesion. (**C**) Sagittal T2 MRI: cervical and thoracic spine that shows continuous signal hyperintensity that involves the spinal cord from cervical to T3–T4 (arrows) in another patient with NMOSD.

**Table 1 biomedicines-11-01898-t001:** Autoimmune diseases associated with COVID-19 Sputnik vaccines *n* = 28.

Syndrome or Disease	Number of Cases	Sex	Age (Years)	Comorbidities	Autoantibodies	1st/2nd Dose	Time to Clinical Manifestations (Days)
Neurological Manifestation							
Guillain Barré Syndrome	6	F: 2 M: 4	32 ±10.5	None: 4 AH:2	ANA: 2, 1:80 (Positive)	1st: 2 2nd: 4	11 (1–30)
Neuromyelitis Optica Spectrum Disorder	2	F: 2	37± 15.6	None	AQP-4: 2 Positive Anti-NMDAr: Negative * Anti- GABA: Negative * Anti-GADAb: Negative *	2nd: 2	7 (5–10)
Myasthenia Gravis	2	M: 1 F: 1	56 ± 20	None	AChR: 2: Positive *	2nd: 2	30
Transverse Myelitis	1	F:1	55	None	ANA: 1:80 (Positive)	2nd	7
Opsoclonus-myoclonus-ataxia syndrome	1	F:1	45	None	ANA: Negative	2nd	24
Chronic Immune Demyelinating Polyneuropathy	1	M: 1	45	AH	ANA: Negative	2nd	45
Hematological Manifestations							
Deep venous thrombosis	6	F:4 M: 2	73 ± 18	AH: 4 Breast cancer: 1	APL: Not done	1st: 3 2nd: 3	32 (10–90)
Multiple venous thrombosis	2	M: 2	54 ± 11	AH	APL: Not done	1st: 2	45 (5–90)
Pulmonary thromboembolism	2	M: 2	85 ± 6	Prostate cancer	APL: Not done	1st: 2	33
Autoimmune hemolytic anemia	2	M: 2	54 ± 11	None	APL: Not done	1st: 2	7
Rheumatological Manifestations							
Dermatomyositis	1	F: 1	40	None	ANA: 1:80 (Positive) Anti-JO1: 10 AU/mL (Negative) Anti-SSA: 12 AU/mL (Negative) Anti-SSB: 15 AU/mL (Negative)	2nd	60
Adult-onset Still’s disease	1	F: 1	20	None	Ferritin: 1000 ng/mL (Positive) RF: 9 IU/mL (Negative) Anti-CCP: 9 AU/mL (Negative) Anti-JO1: 8 AU/mL (Negative) Anti-SSA: 10 AU/mL (Negative) Anti-SSB: 10 AU/mL (Negative)	2nd	60
Endocrinological Manifestations							
Graves’ disease	1	M: 1	28	None	ATSH-R & TPO: Positive * ATG: Negative * Free T3 = 15 pg/mL (2.04–4.1) Free T4 = 4.01 ng/dL (0.93–1.71) Anti TSH-R = 19 UiL(0.1.75) TSI = 340 % Baseline (<140)	2nd	6

F: Female. M: Male. AH: Arterial Hypertension. ANA: Antinuclear antibodies. AQP-4: Anti-aquaporin-4 antibody. Anti-NMDA: anti-N-methyl-D-aspartate. Anti-GABA: anti-glutamic acid decarboxylase antibody. Anti-GADAb: anti-glutamic acid decarboxylase antibodies. AChR: Antibodies against receptors for acetylcholine. APL: Antiphospholipid antibodies. ATSH-R: Anti-TSH receptor antibodies. TSI: Thyroid-stimulating immunoglobulin. TPO: anti-peroxidase antibodies. ATG: antithyroglobulin antibodies. * Our laboratory reports positive or negative results.

**Table 2 biomedicines-11-01898-t002:** COVID-19 Sputnik vaccine side effects.

Author	Sex	Age (Years)	1st/2nd Dose	Time to Clinical Manifestations (Days)	Syndrome or Disease
Shokraee K. et al. [8]	Male	41	1st	20	Reactive arthritis
Ameri M. et al. [9]	Female	40	1st	2	Severe eczema
Herrera-Comoglio R. et al. [10]	Female	24	Not mentioned	7	VITT
Mirmosayyeb O, et al. [41]	(1) Female (2) Male	(1) 27 (2) 58	1st	(1) 3 (2) 10	Bell’s palsy
Mahmoudi N. et al. [43]	Female	34	1st	1	Facial Paresis
Naghashzadeh F. et al. [44]	Male	29	2nd	2	Myocarditis

VITT: Vaccine-induced immune thrombocytopenia and thrombosis.

## Data Availability

Data is available on request from the corresponding author (O.V.-L.) due to privacy restrictions.

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
