# Peer review of "New Onset Autoimmune Diseases after the Sputnik Vaccine"

_biomedicines, 2023, doi:10.3390/biomedicines11071898_

Round 1
Reviewer 1 Report
Dear Authors,
I find the submitted manuscript entitled "New onset autoimmune diseases after the Sputnik vaccine" to be scientifically interesting. However, it lacks important information such as numerical outcomes and statistical analysis. I am curious to know if you conducted any biochemical tests on patient samples or determined the neurological outcomes of individuals exposed to the Sputnik vaccine. Did you perform hematological tests, NETs staining, neutrophil activity analysis, or any imaging scans to support your findings? It appears that you have simply presented the number of patients with different clinical abnormalities. Could you please provide more details on how you tested for the presence of the reported neurological and hematological manifestations? The manuscript is scientifically sound, but it lacks essential facts and figures.
Thank you,
Reviewer
Author Response
Response:
Thank you so much for the review,: Blood cell count and blood chemistry were performed in all patients. For patients with autoimmune rheumatological manifestations, we measured: antinuclear antibodies (ANA), anti-DNA, antibodies extracted from the nucleus (ENA), antineutrophil cytoplasmic antibodies (ANCA), rheumatoid factor (RF), anti-cyclic citrullinated peptide and ferritin. For patients with endocrinological manifestations: thyroid profile, thyroid-stimulating immunoglobulin (TSI), anti-peroxidase antibodies (TPO), antithyroglobulin (ATG), and for neurological patients Antibodies to acetylcholine receptor (AChR), anti- aquaporin-4 (AQP4), antibodies against neuronal surface receptors: anti-N-methyl-D-aspartate (anti NMDA), and Anti-glutamic acid decarboxylase antibody (anti-GABA), anti-glutamic acid decarboxylase (GAD) 65 antibodies were performed in Cerebrospinal fluid studies (CSF). Electromyography was performed on patients who required Brain, cervical, lumbar and thoracic spine MRI were performed too.
This has been included in the methodology Lines 101-116
-Regarding NETs staining and neutrophil activity analysis, we do not carry out these stains or analyses.
-We also added some magnetic resonance images of the central nervous system of NMSOD patients. Figures 1A and 1B.
Since the number of patients is small and heterogenous, we report these findings as descriptives in the result section.
We just describe the patients who met the criteria for a new-onset autoimmune disease with a production of antibodies that fulfilled the specific diagnostic/classification criteria and/or nomenclature for each rheumatic/neurology autoimmune disease within a temporal association (≤90 days) with COVID-19 vaccination. This statement has been included in Line 91-95
Reviewer 2 Report
In this manuscript, Vera-Lastra et al evaluated the development of autoimmune diseases after administration of the Sputnik vaccine. Between March 2021 and December 2022, they observed a total of 123 post-vaccine effects, and more specifically 28 that occurred after administration of the Sputnik vaccine. The most frequent autoimmune manifestations in these patients were Guillain-Barré syndrome and thrombosis. The manuscript is well written and easy to read. It should be of interest for the readers of Biomedicines.
Here are some concerns:
In the title of the Table 1, it is mentioned N=18 (line 131). However, the Results section referred to 28 patients (line 107). Moreover, in the table 1, 29 patients are described. Please, homogenize.
Typos:
Line 31: change SARS-CoV2 by SARS-CoV-2
Line 147: define here VITT, not line 216
English langage fine
Author Response
Thank you so much for the review, we have homogenized that sentence and corrected the typos.
Round 2
Reviewer 1 Report
Response to the Author’s reply:
Response:
Thank you so much for the review,: Blood cell count and blood chemistry were performed in all patients. For patients with autoimmune rheumatological manifestations, we measured: antinuclear antibodies (ANA), anti-DNA, antibodies extracted from the nucleus (ENA), antineutrophil cytoplasmic antibodies (ANCA), rheumatoid factor (RF), anti-cyclic citrullinated peptide and ferritin. For patients with endocrinological manifestations: thyroid profile, thyroid-stimulating immunoglobulin (TSI), anti-peroxidase antibodies (TPO), antithyroglobulin (ATG), and for neurological patients Antibodies to acetylcholine receptor (AChR), anti- aquaporin-4 (AQP4), antibodies against neuronal surface receptors: anti-N-methyl-D-aspartate (anti NMDA), and Anti-glutamic acid decarboxylase antibody (anti-GABA), anti-glutamic acid decarboxylase (GAD) 65 antibodies were performed in Cerebrospinal fluid studies (CSF). Electromyography was performed on patients who required Brain, cervical, lumbar and thoracic spine MRI were performed too.
This has been included in the methodology Lines 101-116
Comment: Where are data? I can’t see any data related to these testing.
-Regarding NETs staining and neutrophil activity analysis, we do not carry out these stains or analyses.
-We also added some magnetic resonance images of the central nervous system of NMSOD patients. Figures 1A and 1B.
Comment: You have added only 1 Brain and 1 spinal cord MRI image. Authors should have presented more images to show validity and robustness of data at least 1 image from every hematological and neurological syndrome bearing patient.
Since the number of patients is small and heterogenous, we report these findings as descriptives in the result section.
We just describe the patients who met the criteria for a new-onset autoimmune disease with a production of antibodies that fulfilled the specific diagnostic/classification criteria and/or nomenclature for each rheumatic/neurology autoimmune disease within a temporal association (≤90 days) with COVID-19 vaccination. This statement has been included in Line 91-95
Comment: Again, the data? I recommend submitting data to show the neurological and hematological manifestations was tested and based on that any conclusion was made. Including this information would improve the transparency and credibility of the study.
Author Response
Dear reviewer, we sincerely appreciate your feedback. We have included the reports of magnetic resonance imaging and anti-aquaporin from another patient. Unfortunately, regarding the thrombotic events, the diagnosis was made clinically, and we do not currently possess the corresponding images. Additionally, we did not perform antibody tests to search for antiphospholipid antibodies. All of these details have been incorporated into the final document.
Furthermore, in the table where we report the cases, we have added the antibodies found in each disease. These are described in the main text, noting that multiple antibodies were tested, but none yielded positive results. We have also attached our patient database for reference. It is important to emphasize that this database is derived from a larger dataset that encompasses all adverse effects associated with different COVID-19 vaccines; however, it is not the focus of the present study.
|
N |
Age |
Sex |
comorbidity |
Vaccine |
Dose |
Time to presentation |
Autoinmune Syndrome |
ANA |
AaChR |
AQP4 |
ATSHR |
Anti TSHR |
|
|
1 |
37 |
M |
AH |
Sputnik |
2 |
32 |
G.Barre |
|
|
|
|
|
|
|
2 |
37 |
M |
AH |
Sputnik |
2 |
2 |
G.Barre |
|
|
|
|
|
|
|
3 |
32 |
M |
None |
Sputnik |
1 |
1 |
G.Barre |
|
|
|
|
|
|
|
4 |
51 |
F |
None |
Sputnik |
2 |
1 |
Myelitis transversa |
+ |
|
|
|
|
|
|
5 |
22 |
F |
None |
Sputnik |
2 |
5 |
NMOSD |
|
|
x |
|
|
|
|
6 |
50 54 |
F M |
NOne |
Sputnik |
2 7
|
10 |
NMSOD AHA |
|
|
x |
|
|
|
|
7 |
28 |
M |
None |
Sputnik |
2 |
6 |
G.Barre |
|
|
|
|
|
|
|
8 |
34 |
F |
None |
Sputnik |
2 |
80 |
NMOSD |
|
|
|
|
|
|
|
9 |
40 |
F |
None |
Sputnik |
2 |
80 |
Derma tomyositis
|
|
|
|
|
|
|
|
10 |
19 |
F |
None |
Sputnik |
2 |
80 |
Still Disease |
|
|
|
|
|
|
|
11 |
33 |
M |
None |
Sputnik |
1 |
1 |
G. Barre |
|
|
|
|
|
|
|
12 |
58 |
M |
None |
Sputnik |
2 |
15 |
TVR |
|
|
|
|
|
|
|
13 |
35 |
F |
Hypotiroidism |
Sputnik |
1 |
60 |
TVM TVP |
|
|
|
|
|
|
|
14 |
72 |
M |
AH, MI |
Sputnik |
2 |
90 |
Miastenia gravis |
|
x |
|
|
|
|
|
15 |
61 |
M |
AH |
Sputnik |
2 |
- |
Chronic demyelinating Polyneuropatia
|
|
|
|
|
|
|
|
16 |
71 |
F |
AH, Choalngitis |
Sputnik |
2 |
30 |
G. Barre |
|
|
|
|
|
|
|
17 |
55 |
F |
AH , DM |
Sputnik |
1 |
7 |
Myasthenia gravis |
x |
|
|
|
|
|
|
18 |
30 28 |
M F |
None None |
Sputnik |
3 2 |
1 6 |
Myasthenia Gravis Graves´disease |
|
|
|
|
|
|
|
19 |
53 |
F |
None |
Sputnik |
2 |
14 |
Opsoclonos myoclonus |
|
|
|
|
|
|
|
20 |
66 |
M |
AH |
Sputnik |
1 |
90 |
TVM |
|
|
|
|
|
|
|
21 |
59 |
M |
Prostae cance |
Sputnik |
1 |
30 |
Embilismo pulmonar |
|
|
|
|
|
|
|
22 |
94 |
M |
None |
Sputnik |
1 |
30 |
TVP |
|
|
|
|
|
|
|
23 |
67 |
M |
AH |
Sputnik |
1 |
10 |
TVP |
|
|
|
|
|
|
|
24 |
89 |
F |
AH |
Sputnik |
1 |
30 |
TVP |
|
|
|
|
|
|
|
25 |
95 |
F |
AH |
Sputnik |
2 |
30 |
TVP |
|
|
|
|
|
|
|
26 |
48 |
F |
Deep venous Thrombosis, cáncer breast |
Sputnik |
1 |
30 |
TVM |
|
|
|
|
|
|
|
27 |
58 |
F |
Parkinson´disease |
Sputnik |
2 |
60 |
Embolismo pulmonar |
|
|
|
|
|
|
|
28 |
76 |
M |
AH |
Sputnik |
1 |
30 |
TVP |
|
|
|
|
|
|
Round 3
Reviewer 1 Report
Dear Authors,
I have carefully reviewed your manuscript and have identified several areas that require attention and revision. Please find below a summary of the issues that need to be addressed:
1. Method Section:
a) For patients with autoimmune rheumatological manifestations, you measured various autoantibodies. Please provide numerical data for all the antibodies tested, not just the ones mentioned in the results section.
b) For patients with endocrinological manifestations, you evaluated specific markers. Include numerical data for all the markers tested.
c) For neurological patients, you tested for several antibodies and performed CSF studies and MRI. Include numerical data for all the antibodies tested and specify the findings from CSF studies and MRI.
I can see only ANA, AQP, AchR, ANA, APL (not done), ferritine, ATCH-R and TSI data. Mention rest of the data in its numerical form and also mention the methods obtained to evaluate such observation.
2. Results Section:
a) Remove repeated images (Fig 2 a and b) and ensure proper figure legends are provided.
b) In Figure 1, the description does not match the actual image. There are yellow dots instead of green and red dots as described by you. Please update the figure and caption accordingly.
c) Provide numerical data for all the autoantibodies mentioned in the method section, not just the ones currently listed in the results section.
d) Table 2 should be removed from the results section as it does not belong there.
3. Discussion Section:
a) The information provided in the discussion section does not adequately justify the results presented in the results section. Ensure that the discussion is closely aligned with the presented data.
4. Figure 1:
a) The methodology for treating mouse frozen muscle sections with Human CSF should be specifically mentioned in the methodology section, not as a figure caption.
b) The significance of this study, including the rationale behind the mentioned experiment, is not discussed in either the results or discussion section. Please address this issue.
Overall, the submitted data appears to be incomplete, and the flow of discussion does not align with the results section. I recommend that you thoroughly revise and address these concerns before resubmitting the manuscript.
Please ensure that the revised manuscript effectively incorporates my suggestions and addresses the mentioned issues.
Author Response
Thank you for your review, attached are the answers to all the suggestions.
